# DVC-SGRL: Adapting MLLMs for Temporally Precise Dense Video Captioning via Semantically Guided Reinforcement Learning

## Abstract

Dense Video Captioning (DVC) aims to localize and describe multiple events within untrimmed videos. While methods using Multimodal Large Language Models (MLLMs) show promise, their ability to precisely localize event boundaries remains a significant limitation. This weakness stems from a reliance on supervised fine-tuning with cross-entropy loss, which frames timestamp prediction as a classification task. In this formulation, the model learns only to match timestamps exactly, with no awareness of how close a prediction is to the ground truth. This limits its ability to interpret time as a continuous signal, hindering accurate event localization. To address this, we introduce DVC-SGRL, a reinforcement learning framework that provides semantically guided temporal supervision, enabling general-purpose MLLMs to be successfully adapted for dense video captioning. Our approach leverages the model's powerful captioning abilities to improve its weaker temporal localization through a novel matching mechanism and corresponding rewards mechanism. Our semantically-guided reward function uses strong matches in caption content to create robust learning signals for refining event boundaries. This "soft alignment" approach, which decouples the evaluation of content and timing, offers far more informative supervision than standard classification losses. Experimental results demonstrate that DVC-SGRL achieves significant improvements in both localization and captioning performance, ultimately reaching state-of-the-art results on YouCook2 and ActivityNet Captions.

## 1 Introduction

Generating rich, temporally grounded descriptions from untrimmed videos is essential for comprehensive machine perception. Dense video captioning tackles this by requiring models to detect and describe multiple events with precise timing and detailed language. It demands joint reasoning over both *what* is happening and *when*, making it a uniquely challenging problem in video understanding.

Earlier DVC (Zhou et al., 2018b; Zhu et al., 2022; Deng et al., 2021) methods typically relied on modular designs that treated localization and captioning as separate tasks. For example, PDVC (Wang et al., 2021) uses parallel heads for event boundary regression and caption generation, which are trained jointly but operate independently (Figure 1a). While effective, separating temporal and semantic modeling restricts the contextual interaction between them, making it challenging for the model to fully integrate temporal structure and linguistic understanding.

The advent of MLLMs (Liu et al., 2023; Bai et al., 2025) has opened new avenues for unified modeling in video understanding. The pioneering approach Vid2Seq(Yang et al., 2023) formulates temporal localization as a sequence generation task, producing event boundaries and descriptions in a single autoregressive output using special time tokens (Figure 1b). While this unified design shows promise, it faces a fundamental limitation: these models rely on supervised fine-tuning with cross-entropy loss, which frames timestamp prediction as a classification task. In this formulation, the model learns only to match timestamps exactly, with no awareness of how close a prediction is to the ground truth. For instance, if the correct timestamp is 1 second, predictions at 2 or 100 seconds are both considered wrong, providing no guidance on how close they are to the correct boundary.

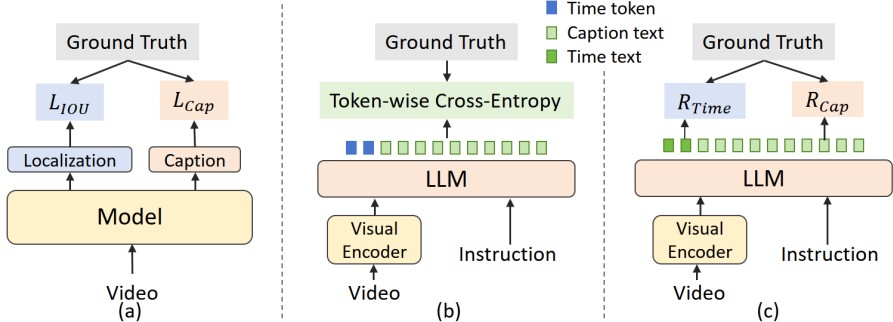

Figure 1: The evolution of dense video captioning paradigms. (a) Traditional modular DVC models like PDVC (Wang et al., 2021) use separate prediction heads for localization and captioning, trained jointly but operating independently, which limits interaction between temporal and semantic modeling. (b) Unified MLLM-based approaches such as Vid2Seq (Yang et al., 2023) and Momentor (Qian et al., 2024) generate interleaved sequences of time tokens and text, but rely on supervised fine-tuning with cross-entropy loss that treats all incorrect timestamps equally, regardless of their distance from the ground truth. (c) Our framework adapts a general-purpose MLLM to DVC by applying reinforcement learning with localization and caption rewards guided by semantic matching, directly optimizing temporal precision and addressing the limitations of prior approaches.

This limits the model's ability to interpret time as a continuous signal and hinders accurate event localization, which remains a critical weakness in MLLM-based DVC approaches.

To address this fundamental challenge, we introduce **DVC-SGRL** (Dense Video Captioning with Semantically Guided Reinforcement Learning), a two-stage training framework that adapts general-purpose MLLMs for temporally precise dense video captioning. We first enable the MLLM to process time by expressing event boundaries in natural language (e.g."01:35 - 01:42"), preserving compatibility with pre-trained models without introducing extra time tokens, modifying the tokenizer, and changing the final token prediction layer. The training proceeds in two stages: an initial supervised fine-tuning phase establishes a foundational understanding of task structure and language, followed by a reinforcement learning stage specifically designed to refine temporal precision.

The core innovation of DVC-SGRL lies in reinforcement learning stage, which employs semantically-guided event matching to leverage the model's strengths in video captioning while addressing its weaknesses in temporal localization. Specifically, predicted events are matched to the ground-truth events with the most semantically similar captions, so that accurate caption content can provide effective supervision for refining their associated timestamps. Temporal localization is optimized using magnitude-aware metrics such as temporal Intersection over Union (tIoU), while caption quality is evaluated separately through semantic similarity. This design allows reliable signals from well-generated captions to guide the refinement of less certain timestamps, providing far more informative supervision than standard cross-entropy losses (Figure 1c). As a result, DVC-SGRL achieves substantial improvements in both event localization and captioning accuracy, attaining state-of-the-art performance on YouCook2 and ActivityNet Captions.

## 2 RELATED WORK

### 2.1 DENSE VIDEO CAPTIONING

Dense Video Captioning (DVC) requires both temporal localization of events and generation of textual descriptions. Early work followed a two-stage *localize-then-describe* pipeline with many hand-crafted components. Recent end-to-end methods such as PDVC (Wang et al., 2021) jointly train localization and captioning via parallel decoding heads. Extensions include E$^2$DVC (Wu et al., 2025) with curriculum learning for rare events, MCCL (Xie et al., 2025) with cyclic co-learning, CM2 (Kim et al., 2024) with memory-augmented cross-modal retrieval, and DIBS (Wu et al., 2024) with pseudo-boundary enrichment for unlabeled data. DiffDVC (Chen et al., 2025) further improves

boundary sensitivity using diffusion models. While effective, these methods still maintain architectural separation between localization and captioning.

A recent shift reformulates DVC as unified sequence generation. Vid2Seq (Yang et al., 2023) extends pretrained language models with temporal tokens to autoregressively output event boundaries and captions in a single sequence, promoting cross-event context and coherence. Streaming DVC (Zhou et al., 2024) adapts this for real-time inference with a *Captioning-in-Advance* strategy and state memory. However, these unified models remain limited by cross-entropy training, which frames timestamp prediction as classification and lacks temporal sensitivity, hindering precise localization.

## 2.2 TEMPORAL UNDERSTANDING IN MLLMs

Recent MLLM advances have introduced architectures and temporal representation techniques that, while not specifically designed for DVC, provide useful insights. TimeChat (Ren et al., 2023) encodes absolute timestamps in visual features for direct temporal expression in text. LITA (Huang et al., 2024) improves localization via relative time tokens and temporal dynamics modeling, trained on specialized benchmarks. Momentor (Qian et al., 2024) achieves fine-grained temporal understanding using large-scale video datasets with segment-level instructions, enabling strong zero-shot reasoning. Qwen2.5-VL (Bai et al., 2025) extends dynamic resolution to the temporal domain, and Time-R1 (Wang et al., 2025) applies reasoning-guided post-training for temporal video grounding, though it addresses only localization, not captioning. While these approaches show strong temporal reasoning, they often require extensive architectural changes, large-scale instruction tuning, or focus on a single temporal aspect. In contrast, our method adapts pretrained MLLMs for joint DVC using lightweight fine-tuning with minimal data and no architectural modifications, efficiently addressing both temporal localization and caption generation without heavy retraining or specialized designs.

## 2.3 REINFORCEMENT LEARNING

Reinforcement learning has proven effective for fine-tuning LLMs beyond supervised learning. A popular approach, Reinforcement Learning from Human Feedback (RLHF)(Ouyang et al., 2022), uses reward models trained on human preferences to optimize non-differentiable objectives, often via Proximal Policy Optimization (PPO)(Schulman et al., 2017). However, PPO can be computationally expensive and unstable. More efficient alternatives have emerged: Direct Preference Optimization (DPO)(Rafailov et al., 2023) directly optimizes policies using pairwise preference data, and Group Relative Policy Optimization (GRPO)(Shao et al., 2024) evaluates multiple generated responses together, producing stronger and more stable learning signals. GRPO has achieved impressive results in applications like DeepSeek-R1 (DeepSeek-AI, 2025) when combined with task-specific rewards. Motivated by these advances, we adopt and extend GRPO for DVC, introducing semantically guided rewards that leverage MLLMs' language understanding to refine both event boundary prediction and caption generation.

## 3 METHOD

We propose a two-stage framework to adapt a pretrained MLLM for DVC. First, supervised fine-tuning establishes a foundational understanding of task structure and language. This is followed by a reinforcement learning stage that refines temporal precision and caption relevance, leveraging GRPO (Shao et al., 2024) with semantic-aware reward functions. An overview of the pipeline is shown in Figure 2.

## 3.1 STAGE 1: SUPERVISED FINE-TUNING

In the first stage, we align a general-purpose MLLM with the DVC task using supervised fine-tuning on annotated datasets. While pretrained MLLMs exhibit promising zero-shot capabilities, SFT serves as a critical adaptation step that teaches the model the structural conventions and language patterns unique to DVC. Specifically, it conditions the model to produce temporally localized descriptions that match the expected output format and granularity.

A central design choice in this stage is how temporal boundaries are represented. Prior approaches often introduce discrete time tokens or custom embeddings (Yang et al., 2023; Huang et al., 2024;

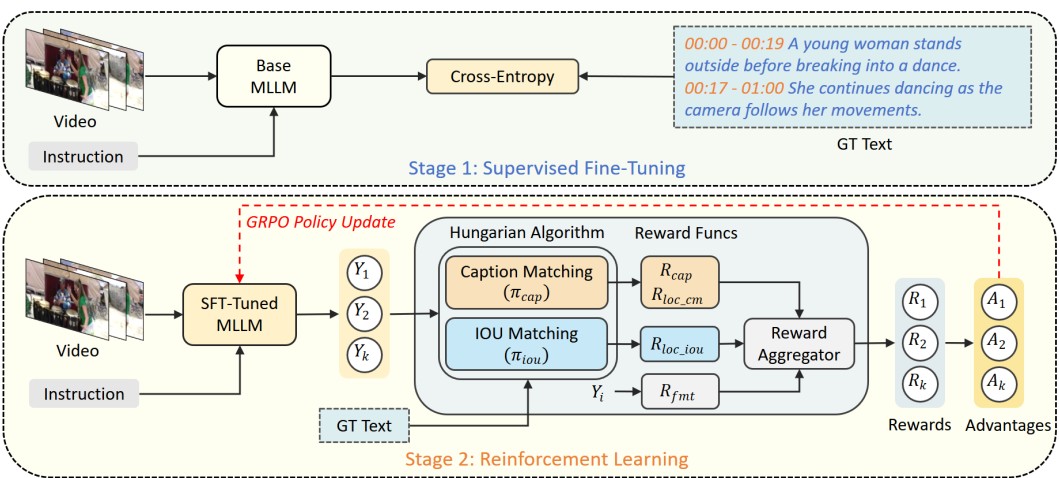

Figure 2: **Two-stage training pipeline of DVC-SGRL.** In the first stage (top), SFT aligns a pre-trained base MLLM with the DVC task using ground-truth events, providing a strong initialization for further RL optimization. In the second stage (bottom), RL with semantically-guided rewards addresses the temporal precision limitation of cross-entropy loss. Multiple sampled predictions are matched with reference events using both caption similarity and temporal IoU via the Hungarian algorithm. The resulting rewards, including caption quality, localization accuracy, and format validity, are combined to compute advantages for stable policy updates using GRPO.

Qian et al., 2024), which can limit generality or require extending the tokenizer. Instead, we leverage the model's existing vocabulary by encoding timestamps as human-readable strings in the `MM:SS` format. For example, 95 seconds is rendered as `"01:35"`, making temporal expressions both natural and parsable. Each event is then represented as:

```
{"start_time": "01:35", "end_time": "01:42", "caption": "A
person is slicing a tomato."}
```

All ground-truth event strings from a single video are concatenated to form the target output sequence $Y_{gt}$, which is then tokenized using the MLLM's native tokenizer. The model is then trained using a standard cross-entropy loss to maximize the likelihood of the target output sequence conditioned on both the visual input $V$ and the task instruction $I$. The SFT objective is defined as:

$$\mathcal{L}_{\text{SFT}}(\theta) = -\sum_{t=1}^{T} \log \pi_\theta(y_t^* \mid y_{<t}^*, V, I), \tag{1}$$

where $y_t^*$ denotes the ground-truth token at time step $t$, and $\pi_\theta$ is the model's output distribution parameterized by $\theta$.

This SFT stage provides the model with a foundational ability to generate temporally grounded captions in the correct format. However, the cross-entropy loss is *temporally insensitive*. It treats any incorrect timestamp token as equally wrong, without reflecting how close or far a prediction is from the ground truth. As a result, the model fails to develop a continuous notion of time, often leading to imprecise boundary predictions. These shortcomings are directly addressed in our second stage through reinforcement learning.

### 3.2 STAGE 2: GRPO-BASED REINFORCEMENT LEARNING

To overcome the limitations of SFT and enhance both temporal localization and caption quality, we introduce a reinforcement learning stage based on GRPO (Shao et al., 2024). This approach is particularly well-suited for our goal of leveraging the MLLM's strong semantic understanding to refine its weaker temporal localization abilities.

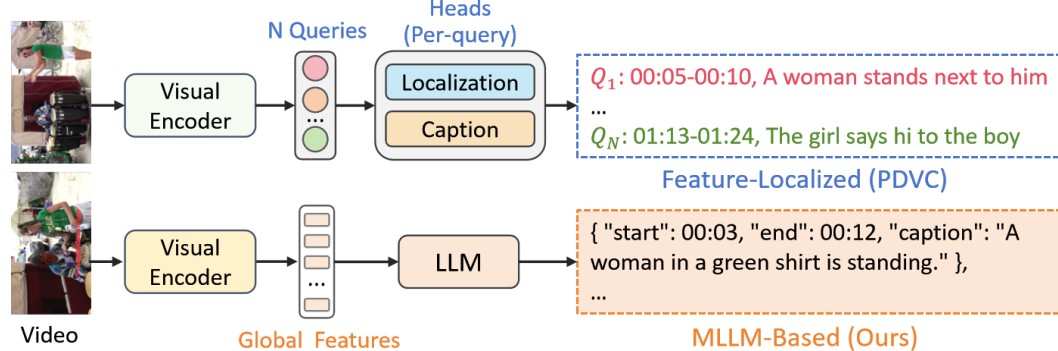

Figure 3: Comparison between PDVC-style feature-localized generation scheme (top) and our MLLM-based approach (bottom). In PDVC, both caption and timestamp are derived from the same query feature, justifying tIoU-based matching. In contrast, our model generates captions and timestamps jointly from global video context, requiring a semantic matching strategy for reward design.

### 3.2.1 GRPO TRAINING FRAMEWORK

As illustrated in Figure 2, the RL stage operates as follows: for a given video input $V$ and instruction $I$, the current policy model $\pi_\theta$ (SFT tuned MLLM) samples a group of $k$ candidate output sequences $\mathcal{G} = \{Y_1, \ldots, Y_k\}$. Each candidate $Y_i$ is a full sequence of predicted events. We then evaluate each candidate using a suite of task-specific reward functions, which are aggregated into a final score $R_i$.

GRPO stabilizes training by avoiding direct optimization on absolute reward values. Instead, it computes the advantage of each candidate relative to the average reward of the group. This advantage, $A_i$, serves as a learning signal. A positive advantage indicates that candidate $Y_i$ is better than the group average, encouraging the policy to increase its probability. Conversely, a negative advantage discourages the generation of outputs like $Y_i$. The policy is updated by maximizing a weighted log-likelihood objective, regularized by a KL divergence term that prevents the policy from deviating too far from the initial SFT tuned model $\pi_{\text{ref}}$:

$$\mathcal{L}_{\text{GRPO}}(\theta) = -\mathbb{E}_{Y_i \sim \pi_\theta} \left[ \sum_{i=1}^{k} A(Y_i) \log \pi_\theta(Y_i) \right] + \beta \cdot \text{KL}(\pi_\theta | \pi_{\text{ref}}), \quad A_i = R_i - \frac{1}{k} \sum_{j=1}^{k} R_j. \quad (2)$$

This framework provides a robust and nuanced learning signal that effectively guides both event localization and caption generation, helping the SFT tuned model improve its performance on dense video captioning.

### 3.2.2 SEMANTICALLY-GUIDED REWARD FUNCTION

**Rationale for Semantic Matching** The design of our reward computation is key to effective reinforcement learning. Prior supervised methods like PDVC (Wang et al., 2021) match predicted events to ground-truth references using temporal IoU (tIoU), forming pairs $\mathcal{M}_{\text{iou}}$, and apply joint supervision to both localization and captioning. This works well for proposal-based architectures, where predicted temporal spans and captions are tightly connected. In our autoregressive MLLM framework (Figure 3), however, captions are generated sequentially from a global context without explicit event proposals. Each caption is therefore not directly grounded in a specific feature span. Consequently, the model often produces semantically correct descriptions with imprecise timing. Using tIoU-based matching in this setting can generate misleading rewards, penalizing the model's language strengths due to weaker temporal alignment and hindering effective learning.

**Semantically-Guided Reward Formulation** To overcome this, we introduce a semantic matching strategy that aligns with the MLLM's capabilities. We first match predicted and reference events based on their caption similarity, creating a new alignment set $\mathcal{M}_{\text{cap}}$. This is achieved by computing pairwise caption similarities with a sentence embedding model, transforming these similarities into costs, and then applying the Hungarian algorithm to find the optimal assignment. This semantic-first

alignment rewards the model for accurate descriptions even when temporal predictions are imperfect, providing a more stable and effective learning signal.

Based on the semantic alignment set $\mathcal{M}_{\text{cap}}$, we define our primary reward components. The **Caption Reward** $R_{\text{cap}}$ encourages fluent descriptions by averaging semantic similarity of matched pairs, while the **Caption-Matched Localization Reward** $R_{\text{loc\_cm}}$ promotes temporal consistency for these semantically-aligned pairs:

$$R_{\text{cap}} = \frac{1}{|\mathcal{M}_{\text{cap}}|} \sum_{(i,j)\in\mathcal{M}_{\text{cap}}} \text{sim}(\hat{c}_i, c_j), \quad R_{\text{loc\_cm}} = \frac{1}{|\mathcal{M}_{\text{cap}}|} \sum_{(i,j)\in\mathcal{M}_{\text{cap}}} \text{tIoU}(\hat{s}_i, s_j) \quad (3)$$

We incorporate two complementary rewards for comprehensive evaluation. To directly measure temporal event segmentation performance, we maintain a traditional IoU-based localization reward $R_{\text{loc\_iou}}$ that assesses localization independently of caption quality using its own matching set $\mathcal{M}_{\text{iou}}$ based purely on maximizing tIoU. Additionally, following recent work on structured output generation such as DeepSeek-R1 (DeepSeek-AI, 2025), we add a format reward $R_{\text{fmt}}$ to guide syntactically valid outputs:

$$R_{\text{loc\_iou}} = \frac{1}{|\mathcal{M}_{\text{iou}}|} \sum_{(i,j)\in\mathcal{M}_{\text{iou}}} \text{tIoU}(\hat{s}_i, s_j), \quad R_{\text{fmt}}(Y) = \begin{cases} 1 & \text{if } \texttt{IsParsable}(Y) \\ 0 & \text{otherwise} \end{cases} \quad (4)$$

The final reward is computed by normalizing each component and combining them as a weighted sum:

$$R_{\text{total}} = \alpha \cdot \text{norm}(R_{\text{cap}}) + \beta \cdot \text{norm}(R_{\text{loc\_cm}}) + \gamma \cdot \text{norm}(R_{\text{loc\_iou}}) + \delta \cdot R_{\text{fmt}} \quad (5)$$

where $\text{norm}(\cdot)$ denotes standardization to $\mathcal{N}(0,1)$ (i.e., subtracting the mean and dividing by the standard deviation across the group), and $\alpha, \beta, \gamma, \delta$ are weighting coefficients. The advantages are then computed from these normalized total rewards for policy gradient optimization.

These four components provide complementary feedback to enhance both localization and captioning performance. Semantic matching improves the caption reward and strengthens the alignment between the predicted time span and its associated description, while localization-based matching using tIoU focuses on assessing the accuracy of event segmentation. Together with format validation, these rewards guide the model to produce temporally precise and semantically coherent outputs.

## 4 EXPERIMENTS

### 4.1 EXPERIMENTAL SETTING

#### 4.1.1 DATASETS AND METRICS

We conduct experiments on two standard dense video captioning benchmarks: ActivityNet Captions (Krishna et al., 2017) and YouCook2 (Zhou et al., 2018a). To evaluate both localization and captioning quality, we use a suite of widely adopted metrics. For captioning, we report METEOR (M) (Banerjee & Lavie, 2005), CIDEr (C) (Vedantam et al., 2015), and SODA-c (S) (Fujita et al., 2020), which together offer a comprehensive assessment of fluency, relevance, and semantic alignment. For event localization, we report Precision (Pre.), Recall (Rec.), and F1 score, computed by evaluating matched predicted and ground-truth event pairs at multiple IoU thresholds (0.3, 0.5, 0.7, and 0.9) and then averaging the resulting metric values across these thresholds.

#### 4.1.2 IMPLEMENTATION DETAILS

We build our method on the 7B version of Qwen2.5-VL (Bai et al., 2025), chosen for its strong baseline performance and its architecture's native support for temporal modeling through dynamic resolution. This enables adaptive processing of multi-frame visual inputs with varying temporal granularity, well-suited for dense video captioning. To preserve its powerful pretrained visual representations, the vision encoder is kept frozen throughout training. Our training process comprises

Table 1: Performance comparison on YouCook2 and ActivityNet. Bold indicates the best performance among all compared methods. Metrics include METEOR (M), CIDEr (C), and SODA-c (S) for captioning quality, and Recall (Rec.) and Precision (Pre.) for event localization. Error ranges for DVC-SGRL are shown below the main values.

| Method | YouCook2 | | | | | ActivityNet | | | | |
|---|---|---|---|---|---|---|---|---|---|---|
| | M | C | S | Rec. | Pre. | M | C | S | Rec. | Pre. |
| **General MLLM** | | | | | | | | | | |
| TimeChat | – | 3.40 | 1.20 | – | – | – | – | – | – | – |
| Momentor | – | – | – | – | – | 4.70 | 14.90 | 2.30 | – | – |
| VTimeLLM | – | – | – | – | – | 6.80 | 27.60 | 5.80 | – | – |
| Qwen2.5-VL | 1.88 | 6.98 | 2.02 | 17.89 | 23.33 | 5.75 | 23.44 | 4.75 | 36.4 | 42.31 |
| **Single Pass DVC models** | | | | | | | | | | |
| PDVC | 4.74 | 22.71 | 4.42 | – | – | 7.96 | 28.96 | 5.44 | 55.42 | 58.07 |
| Vid2Seq | 9.30 | 47.10 | 7.90 | 27.90 | 27.80 | 8.50 | 30.10 | 5.80 | 52.70 | 53.90 |
| CM$^2$ | 6.08 | 31.66 | 5.34 | 24.76 | 33.38 | 8.55 | 33.01 | 6.18 | 53.71 | 56.81 |
| DIBS | 9.41 | 59.35 | 7.97 | 30.80 | **45.13** | 8.25 | 28.85 | 5.35 | 53.02 | 58.39 |
| ILCACM | 4.77 | 13.38 | 3.60 | – | – | 8.48 | 33.42 | 6.08 | – | – |
| DiffDVC | 4.29 | 23.70 | 5.34 | 28.60 | 34.37 | 7.72 | 29.90 | 6.01 | **58.20** | 56.81 |
| E$^2$DVC | 6.11 | 34.26 | 5.39 | 24.36 | 34.75 | 8.57 | 33.63 | 6.13 | 54.67 | 57.70 |
| **DVC-SGRL** | **9.43** | **63.26** | **9.91** | **35.27** | 42.48 | **8.67** | **33.84** | **6.21** | 52.55 | **59.63** |
| | ±0.18 | ±2.16 | ±0.29 | ±1.13 | ±1.82 | ±0.25 | ±0.89 | ±0.13 | ±1.33 | ±1.32 |
| **Multiple Pass DVC models** | | | | | | | | | | |
| Streaming GIT | 3.60 | 15.40 | 3.20 | – | – | 9.00 | 41.20 | 6.60 | – | – |
| Streaming Vid2Seq | 7.10 | 32.90 | 6.00 | – | – | 10.00 | 37.80 | 6.20 | – | – |
| VidChain | 5.60 | 23.80 | 5.60 | – | – | 8.80 | 43.90 | 8.80 | – | – |

two epochs of supervised fine-tuning, followed by two epochs of reinforcement learning. To train a single, unified model for both the YouCook2 and ActivityNet Captions datasets, we construct a combined training set. This includes the full YouCook2 training split (1,333 videos) and a subset of 2,000 videos sampled from ActivityNet Captions, selected to balance the data distribution and mitigate domain bias between the two sources. For reward computation, specifically in $R_{cap}$ and the matching metric of $R_{loc\_cm}$, we compute text embeddings using the Qwen3-Embedding-0.6B (Zhang et al., 2025) model.

## 4.2 COMPARISON WITH STATE-OF-THE-ART METHODS

We compare DVC-SGRL with both general-purpose MLLMs and specialized DVC models on YouCook2 and ActivityNet (Table 1). For reliable evaluation, all scores are reported as mean and standard deviation over three runs. General MLLMs encompass temporal variants such as TimeChat and Momentor, alongside the general-purpose Qwen2.5-VL (7B). While these foundation models demonstrate some video understanding ability, they consistently underperform on both captioning and localization, highlighting the need for task-specific adaptation.

For specialized DVC methods, we evaluate under two paradigms: *single-pass* models, which output all events in a single forward pass, and *multiple-pass* DVC models, which require multiple forward passes to sequentially decode events across the video. DVC-SGRL achieves state-of-the-art captioning performance across both datasets and surpassing strong baselines such as Vid2Seq and DIBS that rely on large-scale video pretraining. For localization, it achieves the highest recall on YouCook2 and the highest precision on ActivityNet. Remarkably, despite operating in the single-pass paradigm, DVC-SGRL even outperforms streaming models on YouCook2, highlighting both efficiency and accuracy. Moreover, our model achieves stronger overall localization, with higher F1 scores than streaming methods (24.1 on YouCook2 and 52.9 on ActivityNet).

The performance differences across datasets stem from both video characteristics and our architectural design. YouCook2 videos are longer on average, making localization more challenging and explaining why ActivityNet achieves higher localization scores. At the same time, YouCook2 benefits from more standardized, action-centric captions focused on cooking steps, whereas ActivityNet involves diverse activities with varying participants and less consistent caption styles. These factors

Table 2: Ablation study on YouCook2 evaluating the effects of different matching strategies and reward components on captioning and localization. We compare IoU-based matching, semantic caption matching, and their combination, with rewards including $R_{\text{cap}}$, $R_{\text{loc\_cm}}$, and $R_{\text{loc\_iou}}$.

| Matching | Rewards | M | C | S | Rec. | Pre. |
|----------|---------|------|-------|------|-------|-------|
| IOU | $R_{\text{loc\_iou}} + R_{\text{cap\_iou}}$ | 8.80 | 58.35 | 9.54 | 33.75 | 40.67 |
| Caption | $R_{\text{cap}}$ | 8.24 | 54.83 | 9.52 | 32.36 | 36.22 |
| Caption | $R_{\text{cap}} + R_{\text{loc\_cm}}$ | 9.35 | 61.98 | 9.62 | 32.46 | 41.70 |
| IOU & Caption | $R_{\text{cap}} + R_{\text{loc\_cm}} + R_{\text{loc\_iou}}$ | **9.43** | **63.26** | **9.91** | **35.27** | **42.48** |

make YouCook2 generally more favorable for caption quality. In addition, our MLLM-based framework predicts event boundaries autoregressively rather than using a dedicated boundary-prediction head as in traditional DVC models. While this design ensures simplicity and tight integration with language modeling, it limits temporal precision on longer videos. Despite these challenges, DVC-SGRL demonstrates strong contextual reasoning, producing coherent captions with accurate grounding. It converges in only four epochs (versus 30 for PDVC and DIBS), requires just 20% of ActivityNet's training data, and maintains robust performance across both datasets with a single generalist model, avoiding dataset-specific tuning. These results highlight how our reward-guided approach effectively adapts a pretrained MLLM into an efficient, high-performing DVC system.

## 4.3 ABLATION ANALYSIS

To evaluate the effectiveness of our approach, we conduct comprehensive ablation studies on the YouCook2 dataset, analyzing the contributions of the proposed semantic matching strategy, reward functions, and two-stage training pipeline to overall model performance.

### 4.3.1 IMPACT OF SEMANTIC MATCHING AND REWARDS

This ablation study examines the effects of different matching strategies and reward mechanisms on YouCook2, as shown in Table 2. Note that $R_{\text{cap\_iou}}$ denotes the caption reward under IoU matching following our proposed reward definition, while the format reward is applied across all experiments but not explicitly listed in the table. The results demonstrate that semantic matching (Caption) significantly outperforms traditional IoU-matching approaches. When comparing IoU matching with $R_{\text{loc\_iou}} + R_{\text{cap\_iou}}$ against caption matching with $R_{\text{cap}} + R_{\text{loc\_cm}}$, we observe that the semantic matching achieves higher scores on most metrics but the Recall, showing its superior ability to generate high-quality captions and maintain precise temporal localization, though with slightly reduced sensitivity in detecting all relevant events.

The analysis of different reward mechanisms reveals their distinct contributions to model performance. The IoU-matching based IoU reward ($R_{\text{loc\_iou}}$) provides rewards based on how well the model can segment the video temporally without being directly involved with matched caption quality, focusing purely on temporal boundary accuracy. Adding the semantic localization reward $R_{\text{loc\_cm}}$ to caption matching substantially improves performance (from 8.24 to 9.35 METEOR, 54.83 to 61.98 CIDEr, 36.22 to 41.70 Pre.), demonstrating the effectiveness of semantic-based temporal reward. The optimal performance is achieved when combining all three reward components: $R_{\text{cap}} + R_{\text{loc\_cm}} + R_{\text{loc\_iou}}$, which yields the highest scores across all metrics. This demonstrates that integrating both semantic matching strategies and complementary reward signals enables more comprehensive learning of both temporal localization and caption generation capabilities, with each reward component contributing to the overall performance improvement.

### 4.3.2 ANALYSIS OF TRAINING STRATEGIES

We conduct an ablation study on the YouCook2 dataset to identify the optimal training strategy, with results shown in Table 3. The findings highlight the distinct and complementary roles of each stage. SFT alone is crucial for establishing a strong baseline, teaching the model the fundamental structure and semantics of the task to achieve respectable performance. In contrast, applying RL from scratch is ineffective. Without a coherent policy initialized through supervised fine-tuning, the model strug-

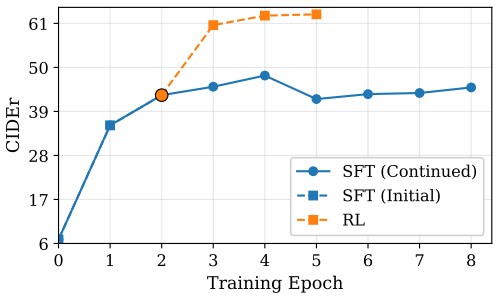 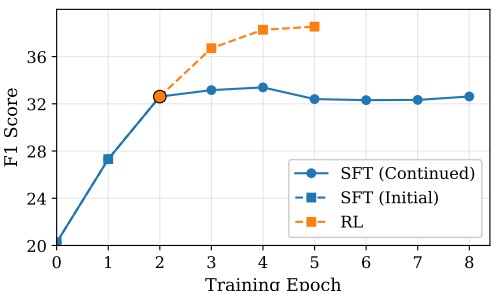

Figure 4: Effect of two-stage training on YouCook2. Left: Captioning performance (CIDEr). Right: Localization performance (F1 Score). Both settings start from the same base model at epoch 0. The blue curve represents the SFT training stage, while the orange curve illustrates the effect of transitioning to RL after two epochs of SFT.

gles to interpret the reward signal effectively, resulting in significantly lower performance in both captioning and localization tasks.

While combining both stages improves performance, the sequence is critical. Our proposed SFT→RL strategy substantially outperforms the reverse order. Although the RL→SFT approach yields the highest recall, our method achieves the best overall balance and secures top performance on nearly all metrics. By first using SFT to build a high-quality, stable policy, the subsequent RL stage can effectively refine this foundation through optimization with task-specific rewards.

Table 3: Impact of different training strategies on the YouCook2 dataset. We compare SFT alone, RL alone, and their combinations with different order. Bold marks the best-performing method for each metric.

| Training Strategy | M | C | S | Rec. | Pre. |
|---|---|---|---|---|---|
| SFT only | 7.72 | 47.96 | 8.47 | 32.00 | 34.92 |
| RL only | 6.53 | 30.48 | 5.04 | 19.37 | 32.59 |
| RL→SFT | 8.07 | 51.12 | 9.22 | **35.36** | 34.67 |
| SFT→RL | **9.43** | **63.26** | **9.91** | 35.27 | **42.48** |

This two-step process results in dramatic gains, achieving state-of-the-art scores across all three captioning metrics and boosting precision by a large margin to 42.48, validating our chosen methodology.

### 4.3.3 SFT Saturation and the Efficacy of RL Refinement

Building on the finding that our two-stage SFT→RL strategy achieves the best overall performance, we examine the limitations of SFT alone and the advantages of the RL stage. As shown in Figure 4, SFT alone (blue curve) plateaus early: most gains in captioning (CIDEr) and localization (F1) occur within the first two epochs, with little improvement thereafter. Introducing RL after two epochs of SFT delivers a clear second-stage boost, as seen in the orange curve, driving substantial gains in both tasks. While SFT provides stable, token-level supervision to establish a strong initial policy, it cannot further optimize model behavior once early learning saturates. RL overcomes this by offering sequence-level, reward-driven feedback that refines both caption quality and temporal localization. Moreover, the RL stage converges rapidly within just two epochs, demonstrating that our two-stage strategy efficiently builds on SFT's foundation to achieve higher performance with minimal computational overhead.

## 5 Conclusion

In this work, we addressed the fundamental limitation of MLLM-based dense video captioning methods that rely on cross-entropy loss, which treats timestamp prediction as classification without temporal distance awareness. We propose DVC-SGRL, a reinforcement learning framework that combines semantically guided matching with complementary rewards, enabling MLLMs' strong

captioning abilities to guide temporal localization through semantic alignment. Our approach maintains full compatibility with pre-trained models while providing richer supervision than standard classification losses. Experiments on YouCook2 and ActivityNet Captions demonstrate state-of-the-art performance, achieving substantial improvements in both event localization and caption quality.

**Reproducibility Statement.** We provide full descriptions of experimental settings, including datasets, training configurations, and evaluation protocols, in Section 4.1. Additional implementation details are included in the Appendix A.2 to ensure clarity and support faithful reproduction of our results. Together, these resources should enable researchers to replicate and extend our work.

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

# A APPENDIX

## A.1 THE USE OF LARGE LANGUAGE MODELS

In this work, large language models were used exclusively to refine the writing and improve readability. They were not involved in the design of methods, execution of experiments, or interpretation of results.

## A.2 IMPLEMENTATION DETAILS

This section provides comprehensive details of our experimental setup across both SFT and RL stages.

### A.2.1 HARDWARE AND SOFTWARE CONFIGURATION

All experiments were conducted with 2 NVIDIA A800 (80GB) GPUs. The software environment included PyTorch 2.6.0, CUDA 12.4, and the Hugging Face TRL library. We employed `Qwen/Qwen2.5-VL-7B-Instruct` (Bai et al., 2025) as the base model..

To enable full-parameter fine-tuning (excluding the vision encoder), we utilized DeepSpeed ZeRO Stage 3 with offloading, configured via the `zero3_offload.json` file. Memory efficiency was optimized through mixed-precision training with `bfloat16` and gradient checkpointing. Additionally, FlashAttention-2 was enabled to accelerate attention computation.

### A.2.2 TRAINING SETUP

Video features were pre-extracted and directly loaded during training to improve computational efficiency. The training prompt used across both stages instructed the model to generate structured outputs in a conversational format, as shown below:

---

**Training Prompt Template**

Localize a series of activity events in the video, output the start and end timestamp for each event, and describe each event with sentences. Provide the result in JSON format with 'mm:ss' format for time depiction as follows:
```
[
    {
        "start_time": "mm:ss",
        "end_time": "mm:ss",
        "description": "Description of the event."
    },
]
```

---

Table 4: GRPO training hyperparameters.

| Hyperparameter | Value |
|---|---|
| Initial Policy ($\pi_{\text{SFT}}$) | SFT-tuned Checkpoint |
| Optimizer | AdamW |
| Training Epochs | 2 |
| Per-Device Batch Size | 1 |
| Gradient Accumulation Steps | 2 |
| Number of Generations | 4 |
| Max Prompt Length | 8192 |
| Max Completion Length | 2048 |
| Freeze Vision Modules | True |
| Weight of Rewards $\alpha, \beta, \gamma, \delta$ | 1 |

Table 5: Performance comparison with and without KL divergence penalty on YouCook2.

| Training Setting | M | C | S | Rec. | Pre. |
|---|---|---|---|---|---|
| w/o KL penalty | 9.16 | 60.44 | 9.82 | 33.38 | 40.52 |
| w/ KL penalty | **9.43** | **63.26** | **9.91** | **35.27** | **42.48** |

Table 6: Impact of model scale and training methodology on YouCook2.

| Training Stage | Size | M | C | S | Rec. | Pre. |
|---|---|---|---|---|---|---|
| SFT only | 3B | 5.21 | 30.27 | 6.46 | 26.46 | 25.41 |
| SFT only | 7B | 6.10 | 43.06 | 7.96 | 32.48 | 34.75 |
| SFT+GRPO | 3B | 7.19 | 43.57 | 7.49 | 25.70 | 35.19 |
| SFT+GRPO | 7B | **9.43** | **63.26** | **9.91** | **35.27** | **42.48** |

**Stage 1 - Supervised Fine-Tuning:** We trained the model using the `SFTTrainer` from the TRL library with standard supervised learning objectives.

**Stage 2 - Group Relative Policy Optimization:** We applied GRPO (Shao et al., 2024) using TRL's `GRPOTrainer`, extended to support multimodal video inputs. Training was initialized from the SFT checkpoint with the vision encoder frozen. For each prompt, the model generated multiple candidate responses evaluated using a custom reward function. The key hyperparameters are summarized in Table 4.

## A.3 ADDITIONAL ABLATION STUDIES

We conduct additional analyses as follows to further assess the robustness of our approach:

### A.3.1 ANALYSIS OF CAPTION SIMILARITY MEASURES

We evaluate the effect of the caption similarity measure, a key component of our semantically-guided reward. It serves two purposes: aligning predicted and ground-truth events, and computing the caption reward. We compare a lexical metric (ROUGE-L) with our embedding-based similarity metric. As shown in Table 7, the choice of similarity measure is especially important when training from scratch. ROUGE-L performs poorly when the model's outputs lack alignment with the target domain, failing to capture semantically relevant matches. This results in weak alignments, sparse or misleading rewards, and a low CIDEr score of 13.48. In contrast, the embedding-based metric captures semantic meaning, enabling more reliable matches and stronger reward signals, which boosts CIDEr to 31.19. Even with SFT initialization, where ROUGE-L benefits from improved outputs, the embedding-based measure still performs better. By recognizing semantically similar captions beyond exact word matches, it offers a richer and more robust learning signal that supports improved performance.

### A.3.2 IMPACT OF FREEZING THE VISION ENCODER

We investigate the effect of fine-tuning the vision encoder and find that keeping it frozen yields consistently better results. As shown in Table 8, the frozen encoder outperforms the trainable version

Table 7: Ablation study on caption similarity measures used in our RL reward, evaluated on YouCook2.

| Sim Measure | SFT Prior | M | C | S | Rec. | Pre. |
|---|---|---|---|---|---|---|
| ROUGE-L | × | 6.44 | 13.48 | 3.96 | 14.55 | 16.97 |
| Text Embedding | × | 6.85 | 31.19 | 5.09 | 32.23 | 19.54 |
| ROUGE-L | ✓ | 8.98 | 60.83 | **10.08** | 33.27 | 40.03 |
| Text Embedding | ✓ | **9.43** | **63.26** | 9.91 | **35.27** | **42.48** |

Table 8: Ablation study on the impact of freezing the vision encoder. Keeping the encoder frozen consistently outperforms fine-tuning, demonstrating the effectiveness of pre-trained visual features and avoiding overfitting on limited data.

| Vision Encoder | M | C | S | Rec. | Pre. |
|---|---|---|---|---|---|
| Trainable | 9.01 | 57.86 | 9.52 | 32.91 | 40.9 |
| Frozen | **9.43** | **63.26** | **9.91** | **35.27** | **42.48** |

across all metrics, including a notable improvement of 4 points in CIDEr. This suggests that the pre-trained vision encoder already provides strong, general-purpose features well-suited to the task. Fine-tuning on limited DVC data risks overfitting and degrades these robust representations. In addition to being more computationally efficient, freezing the encoder proves essential for achieving state-of-the-art performance.

### A.3.3 IMPACT OF KL DIVERGENCE REGULARIZATION

We compare our model trained with KL penalty against a variant without regularization. Table 5 shows that KL regularization improves all metrics, with precision increasing from 40.52% to 42.15%. This confirms that KL regularization prevents policy drift and ensures stable optimization.

### A.3.4 IMPACT OF MODEL SIZE

We compare 3B and 7B versions of Qwen2.5-VL-Instruct. Table 6 shows the 7B model consistently outperforms the 3B variant. After SFT+GRPO, the 7B model achieves METEOR 9.45 vs. 7.19 for 3B. GRPO provides substantial gains at both scales, demonstrating that our pipeline benefits all model sizes while larger models achieve superior performance.

### A.4 QUALITATIVE ANALYSIS

To demonstrate the effectiveness of our DVC-SGRL approach, we present qualitative results on both YouCook2 (Zhou et al., 2018a) and ActivityNet (Krishna et al., 2017) datasets in Figures 5 and 6, respectively. These visualizations illustrate that our method successfully predicts accurate event boundaries while generating rich, descriptive captions that capture the temporal dynamics and content of video segments.

### A.5 QUALITATIVE ANALYSIS ON UNSEEN DOMAIN: NEWS VIDEOS

To assess the generalization capability of DVC-SGRL beyond standard benchmarks, we conducted a qualitative analysis on news videos.

**Ground Truth Generation:** We randomly collected news clips and **manually annotated** the event boundaries to ensure temporal precision. Reference captions were generated by feeding these manually trimmed segments into Qwen2.5-VL-7B, utilizing the strong foundation model to provide high-quality semantic descriptions.

**Analysis:** As shown in Figure 7, DVC-SGRL demonstrates significantly better localization performance than the base model, accurately detecting event transitions in this unseen domain. While the captioning style retains a slight action-centric bias from the training data, the robust temporal grounding confirms that our RL framework successfully instills a generalized, class-agnostic sense of temporal magnitude.

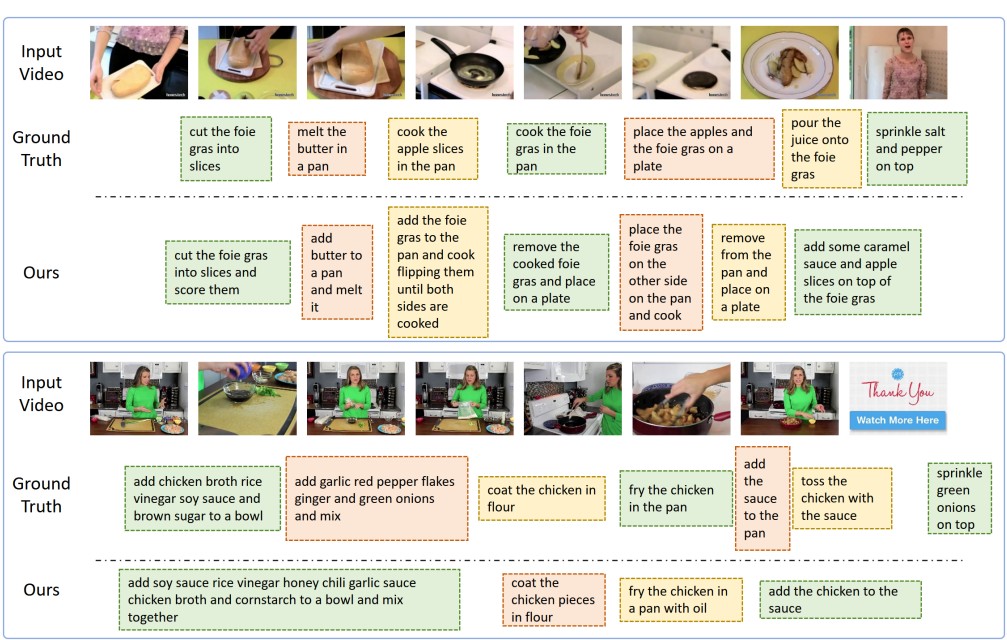

Figure 5: Qualitative results on YouCook2 dataset. Our DVC-SGRL method accurately localizes events and generates detailed descriptions capturing cooking activities and temporal transitions.

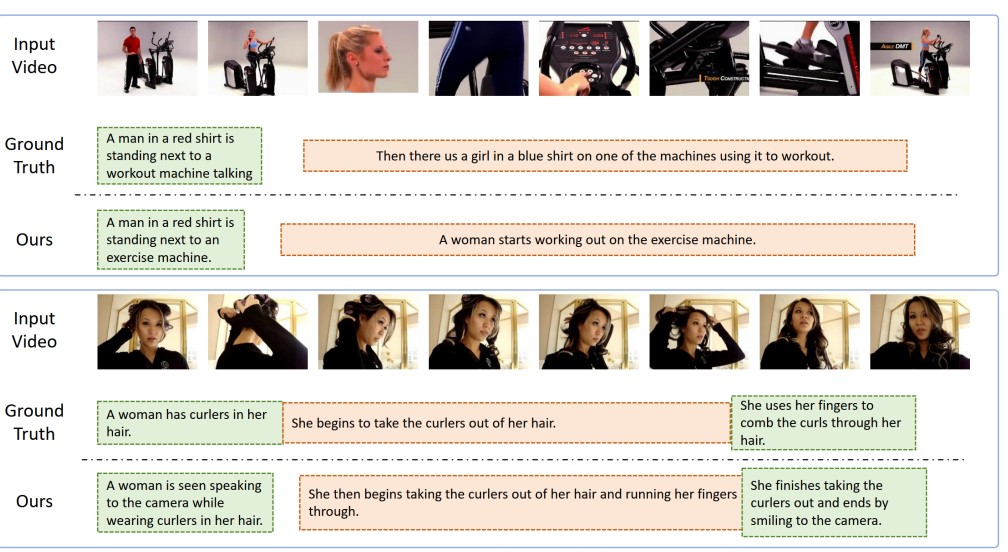

Figure 6: Qualitative results on ActivityNet dataset. The method demonstrates robust performance across diverse activity categories, providing precise temporal localization and comprehensive event descriptions.

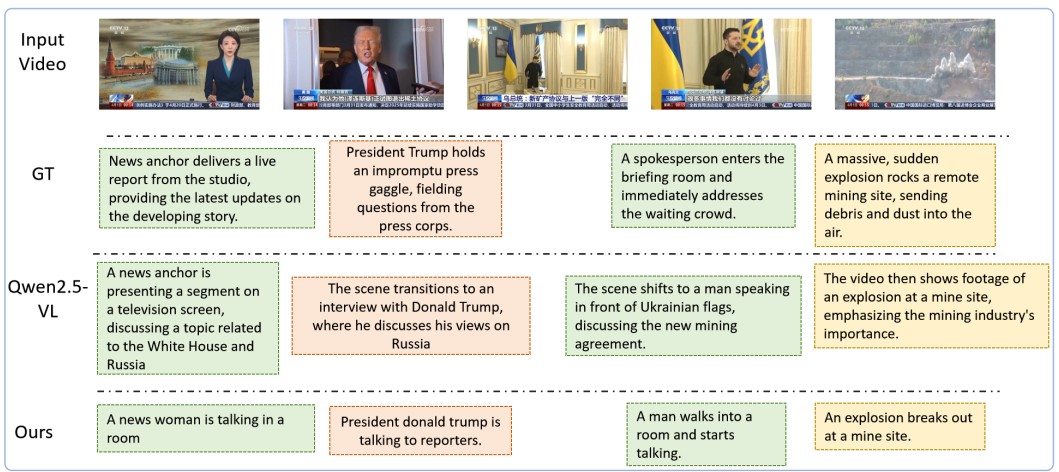

Figure 7: Qualitative results on news video.

