# OpenReview forum: "DVC-SGRL: Adapting MLLMs for Temporally Precise Dense Video Captioning via Semantically Guided Reinforcement Learning"
_ICLR.cc/2026/Conference — Submitted to ICLR 2026_

### Official Review · Reviewer_Ldg9 · 2025-10-31

**Soundness:** 3
**Presentation:** 3
**Contribution:** 2
**Rating:** 4
**Confidence:** 4

**Summary:**

This paper presents DVC-SGRL, a reinforcement learning framework for dense video captioning that adapts multimodal large language models to achieve temporally precise event localization and fluent caption generation. It introduces semantically guided rewards that align predicted and reference events based on caption similarity, allowing strong linguistic understanding to guide boundary refinement.

**Strengths:**

1. **Strong experimental results:** DVC-SGRL achieves superior performance on both YouCook2 and ActivityNet, surpassing prior state-of-the-art methods in both localization and captioning quality.
2. **Sound method design:** The approach effectively addresses the temporal insensitivity of cross-entropy loss through GRPO-based reinforcement learning, providing a well-motivated and efficient solution that avoids architectural modifications or additional temporal tokens.

**Weaknesses:**

1. **Limited novelty:** While temporal sensitivity is indeed an important challenge, prior works [1, 2, 3] have already explored solutions to this issue. The main advancement of DVC-SGRL lies in applying GRPO to dense video captioning, which, although effective, represents an incremental rather than fundamentally novel contribution.
2. **General applicability**: The method is highly tailored toward dense video captioning. Therefore, general capability of the model beyond dense video captinoing might be limited.

**Questions:**

1. Are there specific reason or ablations on choice of $\alpha, \beta, \gamma, \delta$ for reward calculation?
2. Does DVC-SGRL also capable of using speech input as Vid2Seq? Or does it already uses speech input?

---

> ### Author Response · Authors · 2025-11-25
>
> We thank the reviewer for the constructive comments and insightful suggestions.
>
> ---
> **Q1: Concerns regarding limited novelty, specifically that "prior works [1, 2, 3] have already explored solutions to this issue."**
>
> **A1:** We respectfully point out that there appears to be a misunderstanding regarding the references cited as [1, 2, 3] in our manuscript. We have verified our bibliography, where these indices correspond to works that are distinct from our contribution:
>
> * **[1] Qwen2.5-VL :** The technical report for the general-purpose MLLM we employ, not a method for DVC temporal refinement.
> * **[2] METEOR :** A standard text generation evaluation metric, unrelated to temporal localization.
> * **[3] DiffDVC :** A diffusion-based DVC method. While it addresses localization, it relies on a specialized, non-autoregressive diffusion architecture with separate heads,  fundamentally different from adapting autoregressive MLLMs via reinforcement learning.
>
> None of these works represent a prior exploration of **RL-based temporal refinement for MLLMs**, which is the core contribution of our **DVC-SGRL** framework.
>
> Could the reviewer kindly clarify which specific works were intended by references [1, 2, 3]? We would appreciate clarification and would gladly include a detailed comparison.
>
> ---
>
> **Q2: General applicability: The method is highly tailored toward dense video captioning. Therefore, general capability of the model beyond dense video captioning might be limited.**
>
> **A2:** We respectfully point out that **tailoring the method to Dense Video Captioning is the explicit goal and primary contribution of this research**, as stated in the paper's title ("Adapting MLLMs for... Dense Video Captioning") and Abstract.
>
> Specializing the method for DVC is intentional and central to the paper's scope. Our motivation is that general-purpose MLLMs consistently fail on DVC, largely due to cross-entropy training and the lack of temporal magnitude awareness. As shown in **Table 1**, generalist models like TimeChat and Qwen2.5-VL struggle significantly with this task. Thus, specialization is not a limitation but a necessary design choice to address a well-documented gap.
>
> ---
>
> **Q3: Are there specific reasons or ablations on the choice of $\alpha, \beta, \gamma, \delta$ for reward calculation?**
>
> **A3:** Yes. The choice of uniform weighting ($\alpha=\beta=\gamma=\delta=1$) is grounded in our **reward normalization strategy**, and we performed ablations to confirm that this choice is both optimal and robust.
>
> 1. Normalization Ensures Robustness
> As detailed in Equation 5, each reward component is normalized to a standard distribution $\mathcal{N}(0,1)$ within each sampled group before aggregation. This aligns the magnitudes of heterogeneous reward signals, ensuring they contribute comparably to the optimization. As a result, the method is largely insensitive to the exact coefficient values.
>
> 2. Empirical Evidence (Sensitivity Ablation)
> We conducted a sensitivity analysis on YouCook2, systematically increasing individual weights to 2 while keeping others at 1 to evaluate robustness.
>
> **Table R1: Sensitivity Analysis of Reward Coefficients on YouCook2**
>
> | Configuration | METEOR | CIDEr | SODA-c | Recall | Precision |
> | :--- | :---: | :---: | :---: | :---: | :---: |
> | **Default ($\alpha=\beta=\gamma=\delta=1$)** | **9.43** | **63.26** | 9.91 | **35.27** | 42.48 |
> | High Caption Weight ($\alpha=2$) | 9.27 | 62.98 | 9.93 | 34.24 | 39.14 |
> | High Semantic Loc. Weight ($\beta=2$) | 9.06 | 61.82 | **10.05** | 34.54 | **42.59** |
> | High IoU Loc. Weight ($\gamma=2$) | 8.96 | 59.34 | 9.68 | 33.38 | 41.52 |
> | High Format Weight ($\delta=2$) | 8.82 | 58.98 | 9.94 | 33.15 | 39.74 |
> | w/o Normalization | 8.86 | 57.65 | 9.74 | 33.02 | 40.01 |
>
> The results confirm that the **Default (Uniform)** configuration provides the most balanced and superior performance across metrics. Manually biasing weights (e.g., doubling $\gamma$) disrupts the balance established by normalization, leading to drops in specific metrics (e.g., ~4 points in CIDEr). Removing normalization entirely also causes significant performance degradation.
>
> ---
>
> **Q4: Is DVC-SGRL capable of using speech input like Vid2Seq? Or does it already use speech input?**
>
> **A4:** Yes, DVC-SGRL can utilize speech input by appending ASR transcripts to the text instruction prompt. However, we did **not** use speech in training and inference for:
>
> 1. **Task Challenge and Validity:** In YouCook2 and ActivityNet, speech often explicitly describes events. Relying on ASR would shift the task from visual understanding to text summarization, reducing the challenge and failing to evaluate the model’s visual perception capabilities.
>
> 2. **Fair Comparison:** Most standard DVC baselines evaluate performance based solely on visual input. To ensure a fair comparison, we excluded speech modalities.

---

### Official Review · Reviewer_LjyV · 2025-11-01

**Soundness:** 2
**Presentation:** 3
**Contribution:** 2
**Rating:** 4
**Confidence:** 5

**Summary:**

This paper identifies a fundamental weakness in existing multimodal Large Language Model (mLLM) approaches for Dense Video Captioning (DVC): imprecise temporal localization since mLLM is trained with token classification. To address this, the paper proposes DVC-SGRL, a two-stage training framework. Stage 1 performs standard SFT to align a general-purpose MLLM with the task format, notably representing time as natural language strings (e.g., "01:35 - 01:42") to avoid architectural changes. Stage 2, the core innovation, uses reinforcement learning (specifically, GRPO) with a semantically-guided reward function. This function first matches predicted events to ground-truth events based on caption similarity, not temporal overlap. This "soft alignment" allows the MLLM's strong semantic captioning ability to provide a learning signal for its weaker temporal localization, using a reward that combines caption quality, semantically-matched localization, and a separate localization-only score.

**Strengths:**

- The paper is well-written and easy to follow.
- To the best of my knowledge, this is the first paper, which adapts GRPO to a dense video captioning task.
- From the author’s experiments, the proposed DVC-SGRL achieves the best performance compared to other baselines on two benchmarks (ActivityNet and YouCook2). The performance gain seems to be meaningful. In particular, from Table 3, the reinforcement learning-based training strategy shows the performance improvement over other training strategies.

**Weaknesses:**

- More deeper analysis of the semantic matching strategy is required. Compared to existing RL-based mLLMs designed for temporal grounding, the core and original method of this paper is a semantically-guided reward formulation. But, there is a lack of deeper analysis of the semantic matching strategy.
    - First, I wonder why the author applies the Hungarian algorithm to find the optimal assignment. Since the Hungarian algorithm performs one-to-one matching, it may be problematic when the number of predicted captions is different from that of reference captions. It would be better if the paper discussed how to resolve this case.
    - Second, for the Hungarian matching, the paper only uses pairwise caption similarities as a measurement. But, in the DVC task, not only semantic matching but also temporal matching should be considered.
    - Third, I wonder how robust the Hungarian matching algorithm is. It would be better if the author included the analysis concerning the performance of the Hungarian matching algorithm.
    - Forth, I think that employing SODA [1] as a verifiable reward function can play a role in matching rationale. Could you include the experimental results of the model trained with SODA as a verifiable reward function?
- There are some missing baselines such as VTimeLLM, VideoLLaMA2, and VidChain.

[1] Fujita, Soichiro, et al. "Soda: Story oriented dense video captioning evaluation framework." ECCV, 2020.

**Questions:**

The paper seems to apply the same reward value to all the captions in the sequence generated by multimodal LLMs. But, I think that the better way is to apply different reward values to each caption since the caption’s quality is different.

---

> ### Author Response · Authors · 2025-11-25
>
> We thank the reviewer for the constructive comments and insightful suggestions.
>
> ---
>
> **Q1-1: Why apply the Hungarian algorithm, and how are differing numbers of predicted vs. reference captions handled?**
>
> **A1-1:** We use the Hungarian algorithm because it is the standard, verifiable approach for set-based prediction (e.g., PDVC), ensuring a globally optimal assignment that maximizes total semantic similarity.
>
> The algorithm handles mismatches by pairing exactly $K = \\min(N_{pred}, N_{ref})$ events. The adjustment of the event number occurs **automatically depending on the instruction following of the MLLM to cover all non-overlapping events**:
>
> * **Fewer Predictions ($N_{pred} < N_{ref}$):** The Localization Reward ($R_{loc}$) tightens matched events to their boundaries, leaving distinct **"empty durations"** containing valid content. To fulfill the instruction of covering the full video, the model learns to fill these gaps with new events, increasing $N_{pred}$.
> * **More Predictions ($N_{pred} > N_{ref}$):** "Extra" predictions result in **overlaps** with optimized events or redundancy. Since the model targets a coherent, non-repetitive narrative, it learns to prune these conflicting descriptions, decreasing $N_{pred}$.
>
> ---
>
> **Q1-2: Why use only pairwise caption similarity for matching? Should temporal cues also be considered?**
>
> **A1-2:** While combining semantic and temporal cues is common, we prioritize **semantic matching** to accommodate the asymmetric strengths of MLLMs—strong captioning but weak initial localization.
>
> - **Avoiding Misalignment:** MLLMs often produce correct descriptions with drifted timestamps. IoU-first matching would misalign these, penalizing correct language. Semantic-first matching anchors correct pairs, enabling the reward to fix the timestamps.
>
> - **Empirical Support:** Table 2 shows IoU-only matching (Row 1) significantly underperforms semantic matching (Row 3): **58.35 → 61.98 CIDEr**.
>
> - **Complementary Use:** We do not discard temporal matching; we use a parallel IoU branch to compute the auxiliary IoU reward ($R_{loc\\_iou}$). As shown in our full method, this supplementary signal yields the best overall performance (63.26 CIDEr).
>
> ---
>
> **Q1-3: How robust is the Hungarian algorithm?**
>
> **A1-3:** To demonstrate the robustness of the Hungarian algorithm over simpler alternatives, we compared it against a **Greedy Matching** baseline. As shown below, Hungarian consistently performs better.
>
> **Table R3: Matching Algorithm Robustness (YouCook2)**
>
> | Matching Algorithm | METEOR | CIDEr | SODA-c | Recall | Precision |
> | :--- | :---: | :---: | :---: | :---: | :---: |
> | Greedy Matching | 9.11 | 60.11 | 9.84 | 33.50 | 41.29 |
> | **Hungarian Matching (Ours)** | **9.43** | **63.26** | **9.91** | **35.27** | **42.48** |
>
> ---
>
> **Q1-4: Can you provide results using SODA as a reward?**
>
> **A1-4:** Yes. We replaced our decoupled rewards with a single SODA reward. Results are shown below.
>
> **Table R4: SODA Reward vs. Decoupled Reward (YouCook2)**
>
> | Reward Strategy | METEOR | CIDEr | SODA-c | Recall | Precision |
> | :--- | :---: | :---: | :---: | :---: | :---: |
> | SODA Reward | 8.99 | 59.48 | 9.90 | 35.08 | 40.90 |
> | **Ours (Decoupled)** | **9.43** | **63.26** | **9.91** | **35.27** | **42.48** |
>
> Our method separates **Semantic Matching** and **Temporal IoU**, while SODA aggregates both into a single scalar. Inaccurate temporal boundaries can inject noise into the SODA score computation during early RL, which may explain its weaker performance.
>
> ---
>
> **Q2: Missing baselines such as VTimeLLM, VideoLLaMA2, and VidChain.**
>
> **A2:** We have updated Table 1 to include **VTimeLLM** and **VidChain**. We exclude **VideoLLaMA2** because it has not been officially evaluated on the DVC task.
>
> ---
>
> **Q3: Would event-level rewards (per-event) perform better than sequence-level rewards?**
>
> **A3:** We implemented an **Event-Level** reward scheme. Results are shown below.
>
> **Table R5: Reward Granularity (YouCook2)**
>
> | Reward Granularity | METEOR | CIDEr | SODA-c | Recall | Precision |
> | :--- | :---: | :---: | :---: | :---: | :---: |
> | Event-Level | 8.96 | 59.34 | 9.68 | 32.99 | 40.95 |
> | **Sequence-Level (Ours)** | **9.43** | **63.26** | **9.91** | **35.27** | **42.48** |
>
> Since different generations produce different numbers of events, computing advantages between “corresponding” events is inherently difficult. As a result, we compute the event-level advantage across all events from all generations, but this broad aggregation introduces substantial statistical noise into the estimation. This noise likely explains why the event-level approach underperforms compared to the holistic sequence-level reward.

---

### Official Review · Reviewer_37Wb · 2025-11-01

**Soundness:** 3
**Presentation:** 3
**Contribution:** 3
**Rating:** 6
**Confidence:** 3

**Summary:**

The paper highlights a limitation of standard cross-entropy training in dense video captioning: timestamps are treated as discrete labels, so the model cannot distinguish between predictions that are slightly off versus significantly incorrect.
To address this, the authors introduce DVC-SGRL, a two-stage training pipeline.
In the first stage, supervised fine-tuning teaches the model the structural format and linguistic patterns required for DVC.
In the second stage, GRPO-based reinforcement learning incorporates the proposed semantically guided matching strategy.

**Strengths:**

- The authors propose DVC-SGRL, a two-stage training pipeline consisting of supervised fine-tuning followed by reinforcement learning. They show that this ordering (SFT → RL) yields the best performance compared to other training sequences.
- They design a composite reward with four components: caption reward, caption-matched localization reward, traditional IoU-based localization reward, and a format reward.
- They adopt human-readable timestamps to avoid relying on special time tokens during supervised training.
- The method achieves strong results on YouCook2 and ActivityNet, outperforming baseline models.
- The paper includes comprehensive ablation studies that demonstrate the effectiveness of the approach across multiple design choices.

**Weaknesses:**

- How sensitive is the method to the weighting coefficients in the reward function?
- How effective is the use of human-readable timestamps? Although the authors list this as a contribution, there is no analysis demonstrating its impact on performance.
- How does the number of training epochs for the SFT and RL stages affect the final performance?

**Questions:**

- In Table 3, the Recall value is incorrectly bolded.

---

> ### Author Response · Authors · 2025-11-25
>
> We thank the reviewer for the constructive comments and insightful suggestions.
>
> ---
> **Q1: How sensitive is the method to the weighting coefficients in the reward function?**
>
> **A1:** Our method is robust to variations in reward weights, thanks to our reward normalization strategy.
>
> 1. Normalization Ensures Robustness
> As detailed in Equation 5, each reward component is normalized to a standard distribution $\mathcal{N}(0,1)$ within each sampled group before aggregation. This aligns the magnitudes of different reward signals, ensuring they contribute to the optimization landscape at a comparable level. As a result, the method is largely insensitive to the exact coefficient values.
>
> 2. Empirical Evidence (Sensitivity Ablation)
> We conducted a sensitivity study on YouCook2, doubling each weight individually to test robustness.
>
> | Setting | METEOR | CIDEr | SODA-c | Recall | Precision |
> | :--- | :---: | :---: | :---: | :---: | :---: |
> | **Default ($\alpha=\beta=\gamma=\delta=1$)** | **9.43** | **63.26** | 9.91 | **35.27** | 42.48 |
> | Caption 2 ($\alpha=2$) | 9.27 | 62.98 | 9.93 | 34.24 | 39.14 |
> | Semantic Loc. 2 ($\beta=2$) | 9.06 | 61.82 | **10.05** | 34.54 | **42.59** |
> | IoU Loc. 2 ($\gamma=2$) | 8.96 | 59.34 | 9.68 | 33.38 | 41.52 |
> | Format 2 ($\delta=2$) | 8.82 | 58.98 | 9.94 | 33.15 | 39.74 |
> | w/o Norm | 8.86 | 57.65 | 9.74 | 33.02 | 40.01 |
>
> Default uniform weights provide the most balanced performance. Biasing individual weights disrupts metric balance, and removing normalization leads to substantial drops (e.g., ~4 points in CIDEr). This confirms that normalization effectively centers the optimal operating point.
>
> ---
>
> **Q2: How effective is the use of human-readable timestamps?**
>
> **A2:** We adopted the "MM:SS" format to maximize compatibility with the pre-trained MLLM and avoid the structural risks of introducing new tokens. To demonstrate its effectiveness, we compared it against a rigorously designed baseline using raw numerical inputs.
>
> **1. Practical and Theoretical Rationale**
> Adding discrete time tokens may be straightforward like Vid2Seq, but it requires modifying the tokenizer and embedding layers, which is non-trivial and risks degrading pre-trained representations. To evaluate the advantage of using human-readable timestamps, we compared two settings:
>
> * **Human-Readable ("MM:SS"):** Our proposed method. It leverages the model's strong pre-trained exposure to standard time formats found in video subtitles and transcripts.
> * **Baseline - Raw Seconds (e.g., "95"):** We established this as a **theoretically strong baseline** to proxy for discrete time tokens. Unlike a newly introduced token (e.g., `<time_95>`) which would be semantically void at initialization, the MLLM has already learned the numerical meaning and ordering of "95" during pre-training. Therefore, "Raw Seconds" represents a more competitive starting point than uninitialized tokens; if "MM:SS" outperforms this, it confirms the superiority of the format itself.
>
> **2. Empirical Validation**
> We conducted an ablation on YouCook2 to compare our method against this baseline. As shown in **Table R2**, the human-readable format outperforms the raw seconds representation across most metrics.
>
> **Table R2: Ablation on Timestamp Formats (YouCook2)**
>
> | Timestamp Format | METEOR | CIDEr | SODA-c | Recall | Precision |
> | :--- | :---: | :---: | :---: | :---: | :---: |
> | Baseline (Raw Seconds) | 9.23 | 61.98 | **10.13** | 34.36 | 41.35 |
> | **Human-Readable ("MM:SS")** | **9.43** | **63.26** | 9.91 | **35.27** | **42.48** |
>
> The results indicate that the "MM:SS" format is better aligned with the MLLM's internal representation of time than raw integers, yielding higher Caption Quality (+1.28 CIDEr) and Localization Precision (+1.13%).
>
> ---
>
> **Q3: How does the number of training epochs for SFT and RL affect performance?**
>
> **A3:** We analyzed training duration in Section 4.3.3 and found distinct convergence behaviors:
>
> * **SFT Saturation:** Most gains in captioning and localization occur within the first two epochs; further SFT yields diminishing returns due to cross-entropy’s lack of magnitude awareness.
> * **RL Refinement:** Switching to RL after two SFT epochs unlocks substantial improvements. The RL stage converges within two epochs, efficiently refining the initial policy.
>
> Based on these results, we use **2 epochs SFT + 2 epochs RL**, achieving strong performance with computational efficiency.
>
> **Q4: In Table 3, the Recall value is incorrectly bolded.**
>
> **A4:** We thank the reviewer for carefully checking our results. We have corrected the bolding in Table 3 in the revised manuscript to accurately reflect the best-performing method.

---

> > ### Comment · Reviewer_37Wb · 2025-11-27
> >
> > Thank you for your detailed response.
> >
> > Regarding Q2, have you also experimented with introducing new time tokens? Additionally, I’m not sure how generalizable this formatting benefit is. It seems more like a design choice that could vary depending on the model rather than a robust mechanism for all models.

---

> ### Author Response · Authors · 2025-11-28
>
> We thank the reviewer for the thoughtful follow-up regarding our timestamp design choices.
>
> **Q2 Follow-up: Have you also experimented with introducing new time tokens? How generalizable is this formatting benefit?**
>
> **A2 (Follow-up):** Yes, we also experimented with introducing new time tokens like Vid2Seq and present the results below.
>
> **1. Experimental Results.**
>
> As shown in the table below, introducing new time tokens results in the lowest performance among all settings, performing even worse than the "Raw Seconds" baseline.
>
> **Table R2-Followup: Comparison of Timestamp Representations (YouCook2)**
>
> | Timestamp Format | METEOR | CIDEr | SODA-c | Recall | Precision |
> | :--- | :---: | :---: | :---: | :---: | :---: |
> | New Time Tokens | 8.75 | 58.42 | 9.45 | 31.80 | 38.55 |
> | Raw Seconds (Baseline) | 9.23 | 61.98 | **10.13** | 34.36 | 41.35 |
> | **Human-Readable (Ours)** | **9.43** | **63.26** | 9.91 | **35.27** | **42.48** |
>
> **2. Explanation: The Pre-training vs. Fine-tuning Gap**.
>
> The poor performance of new time tokens is expected given our training paradigm.
> * **Vid2Seq (Pre-training):** Methods like Vid2Seq work with discrete time tokens because they are **pre-trained** on millions of video-text pairs. This massive scale allows the model to learn the semantic embedding of new time token from scratch.
> * **DVC-SGRL (Fine-tuning):** Our framework is an **adaptation** method. We fine-tune the model for only **4 epochs** (2 SFT + 2 RL) on the target dataset (YouCook2/ActivityNet). This limited duration is insufficient for the model to learn meaningful representations for hundreds of randomly initialized tokens.
> * **Why "MM:SS" Wins:** In a fine-tuning regime, leveraging the **prior knowledge** of the MLLM is crucial. The model already understands the semantic meaning of "01:30" (or even "90") from its vast text pre-training. Introducing new tokens essentially discards this prior, forcing the model to learn a new "language" of time with insufficient data.
>
> **3. Generalizability**
> Regarding generalizability, we argue that the "MM:SS" format is actually **more generalizable** for adapting modern MLLMs than custom tokens.
> * **Foundation Model Nature:** Most current state-of-the-art MLLMs (e.g., Qwen-VL, LLaVA) are initialized from Large Language Models trained on diverse web text, which naturally includes human-readable timestamps (transcripts, subtitles).
> * **Robustness:** Using "MM:SS" aligns with the inherent text-processing nature of these foundation models. In contrast, adding special tokens is the *model-specific* design choice, as it introduces architectural dependencies that vary between tokenizers and require model-specific pre-training recipes to work effectively. Thus, for *adapting* general-purpose MLLMs to video tasks without expensive pre-training, text-based timestamps are the robust solution.

---

### Official Review · Reviewer_jAFC · 2025-11-04

**Soundness:** 3
**Presentation:** 3
**Contribution:** 2
**Rating:** 4
**Confidence:** 5

**Summary:**

Dense Video Captioning (DVC) requires models to both localize events in untrimmed videos and describe them. While VLMs are promising for this task, they struggle with precise event localization. This stems from their reliance on supervised fine-tuning (SFT) which treats timestamp prediction as a classification task. In this formulation, all incorrect timestamps are treated equally regardless of their proximity to the ground truth, preventing the model from interpreting time as a continuous signal.

The paper introduce DVC-SGRL, a two-stage training framework designed to adapt general-purpose VLMs for precise DVC. The model is taught to express event boundaries using natural language strings, rather than specialized time tokens, maintaining compatibility with pretrained models. Then uses an IoU based reward function to further train it. The model is evaluated on YouCook2 and Acitivtynet.

**Strengths:**

The proposed approach is different from prior works and the experiments show it is beneficial. However, prior works treating timestamps as text are common, such as Vid2Seq.

The approach maintains full compatibility with pre-trained MLLMs. It does not require architectural modifications, which is nice.

**Weaknesses:**

The novelty is a bit limited, as it is mostly just a new reward function for dense captioning tasks. It is further questionable how meaningful the ground truth timestamps are, as historically there has always been disagreement and ambiguity in the timestamps of actions from human annotators.

The reward function is also a bit concerning. The implementation details reveal that all weighting coefficients (α,β,γ,δ) are set to 1. While simple, it is highly improbable that a uniform weighting is optimal across different datasets with varying densities of events (e.g., cooking steps in YouCook2 vs. sparse activities in ActivityNet). It is also unclear how important each of the components of the reward function are. The ablation in table 2 doesn't have huge differences between the settings, so it isn't clear if they are statistically significant.

The authors admit that their autoregressive design for predicting event boundaries "limits temporal precision on longer videos". Representing time purely as text tokens ("MM:SS") in a single sequence can lead to drifting errors or context window issues in very long untrimmed videos with high event density.

The paper claims to train a "single, unified model" that avoids dataset-specific tuning. However, this "generalist" model is only trained on a combination of two datasets: YouCook2 and a subset of ActivityNet . These are both relatively standard, activity-centric datasets, and the only datasets the paper is evaluated on. A true "generalist" DVC model should be robust to vastly different video domains

**Questions:**

See weaknesses.

---

> ### Author Response · Authors · 2025-11-25
>
> We thank the reviewer for the constructive comments and insightful suggestions.
>
> ---
> **Q1: Concerns regarding limited novelty and ground truth timestamp ambiguity.**
>
> **A1:**  We respectfully disagree that our work lacks novelty. Our framework addresses a fundamental bottleneck in MLLM-based video understanding.
>
> 1. Novelty: Resolving the "Classification vs. Regression" Conflict
> Our contribution goes beyond a reward function. It introduces a paradigm shift in how MLLMs are trained for continuous temporal interpretation.
>
> * **Cross-Entropy Bottleneck:** Existing MLLMs (e.g., Vid2Seq) use supervised fine-tuning with cross-entropy loss, treating timestamp prediction as classification. Predicting "00:02" for a "00:01" label is penalized as heavily as "00:50," ignoring temporal magnitude and preventing learning of continuous time.
> * **Semantically Guided Framework:** DVC-SGRL is the first framework to adapt general-purpose MLLMs for DVC using reinforcement learning to inject magnitude awareness. The Semantic Matching leverages the model’s strongest modality, language, to guide its weakest modality, temporal localization, thereby overcoming poor initial alignments.
>
> 2. Ground Truth Ambiguity
> We acknowledge that human annotation contains ambiguity. However, all baselines (PDVC, Vid2Seq, DiffDVC) are evaluated against the same labels. DVC-SGRL’s significant performance gains demonstrate that it better captures temporal structure despite annotation ambiguity.
>
> ---
> **Q2: Concerns about uniform reward weights ($\alpha=\beta=\gamma=\delta=1$) and the contribution of each reward component.**
> **A2:** Uniform weighting is a deliberate design choice enabled by our **normalization strategy**, not oversimplification.
>
> 1. Normalization Ensures Comparable Magnitudes
> All reward components are normalized to $\mathcal{N}(0,1)$ within each sampled group (Equation 5), ensuring they contribute at a similar magnitude to the learning signal regardless of dataset characteristics.
>
> 2. Sensitivity Ablation
> We tested individual weight increases to 2 while keeping others at 1:
>
> | Setting | METEOR | CIDEr | SODA-c | Recall | Precision |
> | :--- | :---: | :---: | :---: | :---: | :---: |
> | **Default (1,1,1,1)** | **9.43** | **63.26** | 9.91 | **35.27** | 42.48 |
> | Caption 2 | 9.27 | 62.98 | 9.93 | 34.24 | 39.14 |
> | Semantic Loc. 2 | 9.06 | 61.82 | **10.05** | 34.54 | **42.59** |
> | IoU Loc. 2 | 8.96 | 59.34 | 9.68 | 33.38 | 41.52 |
> | Format 2 | 8.82 | 58.98 | 9.94 | 33.15 | 39.74 |
> | w/o Norm | 8.86 | 57.65 | 9.74 | 33.02 | 40.01 |
>
> Default uniform weights provide the most balanced performance. Manually biasing weights disrupts this balance, confirming that normalization centers the optimal operating point near uniform weighting. Notably, removing normalization yields the worst results, further demonstrating that normalization is essential for stable and effective learning.
>
> 3. Reward Component Significance
> Table 2 in the main paper investigates the impact of *mechanistically* adding or removing different matching strategies and reward functions.
> * Semantic matching ($R_{cap} + R_{loc\\_cm}$) boosts CIDEr from 58.35 → 61.98.
> * IoU reward ($R_{loc\\_iou}$) recovers Localization Precision from 41.70 → 42.48.
>
> These results show each component is essential and their combination empirically optimal.
>
> ---
>
> **Q3: Concerns that autoregressive "MM:SS" timestamps may introduce drifting errors.**
>
> **A3:** We explicitly acknowledge that autoregressive generation can suffer from drift in long videos. However, mitigating this specific vulnerability is the **primary motivation** for DVC-SGRL.
>
> ---
>
> **Q4: Concerns about "single, unified model" and "generalist" claims given training on only YouCook2 and ActivityNet.**
>
> **A4:**
>
> 1. Unified vs. Generalist
>
> “Single, unified model” refers to using one set of weights across multiple benchmarks, unlike standard DVC approaches that train separate models for each dataset. We do not claim a universal generalist model; the general models in our paper are MLLMs, including TimeChat and Qwen2.5-VL, used for comparison.
>
> 2. Standard Benchmarks
> YouCook2 and ActivityNet are the standard benchmarks for DVC. We are not aware of widely used DVC benchmarks that are not activity-centric. Are there any such DVC datasets you would recommend for evaluation?
>
> 3. Generalization to New Domains (News Videos)
> We tested on **news videos**, very different from cooking/daily activities. As in **Appendix A.5**, with manually labeled event boundaries and reference captions from **Qwen2.5-VL-7B**, DVC-SGRL:
> * Detects segment transitions accurately, outperforming the base model without fine-tuning.
> * Captions retain some action-focused style from training, yet **temporal grounding is precise**.
> This indicates RL-based alignment gives the model a **class-agnostic sense of event duration**, rather than merely fitting the training data.

---

### Meta-Review · Area_Chair_NNVy · 2026-01-07

**Summary:**

The paper proposes a reinforcement-learning-based fine-tuning framework to adapt multimodal large language models for dense video captioning, primarily improving event boundary localization through semantically guided temporal supervision.

The initial reviews nearly reached a consensus that the paper falls marginally below the acceptance threshold, with scores of three "Marginally below the acceptance threshold" (4) and one "Marginally above the acceptance threshold" (6).

The major concerns raised by the reviewers are as follows:

1. Limited novelty. The proposed method is merely an adoption of GRPO to dense video captioning, rather than a fundamentally new technical contribution.
2. Limited general applicability. The proposed techniques are primarily tailored to dense video captioning, and the time-formatting choice "MM:SS," while interesting, largely reflects manual prompt engineering rather than a generalizable mechanism.
3. More in-depth analysis and justification of Hungarian matching were requested.
4. Sensitivity analyses of hyperparameters were requested.

**Reviewer Concerns:**

The authors adequately addressed the issues related to Hungarian matching and sensitivity analyses of hyperparameters through additional experimental results. However, as pointed out by a reviewer, concerns related to the limited novelty and technical contributions were not fully resolved in the rebuttal. In particular, the first application or adoption of an existing method to a problem does not, by itself, constitute sufficient novelty or a substantive contribution.

**Reviewer Scores:**

The initial reviews indicate that this paper lies at the borderline or slightly below the acceptance threshold. The authors’ rebuttal effectively clarifies several technical details and provides additional empirical analyses to justify key design choices (e.g., Hungarian matching and timestamp formatting) as well as hyperparameter settings. However, the more substantive concerns such as the limited novelty and technical contribution of the work, remain insufficiently addressed. In particular, the paper primarily represents an early or straightforward application of a recent technique GRPO, which does not sufficiently support its overall merit at this venue. Therefore, I recommend rejection.

---

### Decision · Program_Chairs · 2026-01-26

Reject